# Zeolitic Imidazolate Frameworks Serve as an Interface Layer for Designing Bifunctional Bone Scaffolds with Antibacterial and Osteogenic Performance

**DOI:** 10.3390/nano13212828

**Published:** 2023-10-25

**Authors:** Jingxi Huang, Chen Cheng, Youwen Yang, Jun Zan, Cijun Shuai

**Affiliations:** 1Institute of Additive Manufacturing, Jiangxi University of Science and Technology, Nanchang 330013, Chinayangyouwen@csu.edu.cn (Y.Y.); 2Department of Biomedical Engineering, The Hong Kong Polytechnic University, Hong Kong 999077, China; 3State Key Laboratory of Precision Manufacturing for Extreme Service Performance, College of Mechanical and Electrical Engineering, Central South University, Changsha 410083, China; 4College of Mechanical Engineering, Xinjiang University, Urumqi 830017, China

**Keywords:** bone scaffold, polylactic acid (PLLA), zeolitic imidazolate frameworks (ZIF-8), antibacterial activity, osteogenic performance

## Abstract

The integration of hydroxyapatite (HA) with broad-spectrum bactericidal nano-silver within biopolymer-based bone scaffolds not only promotes new bone growth, but also effectively prevents bacterial infections. However, there are problems such as a poor interface compatibility and easy agglomeration. In this project, zeolitic imidazolate frameworks (ZIF-8) were grown in situ on nano-HA to construct a core–shell structure, and silver was loaded into the ZIF-8 shell through ion exchange. Finally, the core–shell structure (HA@Ag) was composited with polylactic acid (PLLA) to prepare bone scaffolds. In this case, the metal zinc ions of ZIF-8 could form ionic bonds with the phosphate groups of HA by replacing calcium ions, and the imidazole ligands of ZIF-8 could form hydrogen bonds with the carboxyl groups of the PLLA, thus enhancing the interface compatibility between the biopolymers and ceramics. Additionally, the frame structure of MOFs enabled controlling the release of silver ions to achieve a long-term antibacterial performance. The test results showed that the HA@Ag nanoparticles endowed the scaffold with good antibacterial and osteogenic activity. Significantly, the HA@Ag naoaprticle exhibited a good interfacial compatibility with the PLLA matrix and could be relatively evenly dispersed within the matrix. Moreover, the HA@ZIF-8 also effectively enhanced the mechanical strength and degradation rate of the PLLA scaffold.

## 1. Introduction

Bone defects are one of the most common tissue damages, resulting from traumas, infections, tumor resections, and reconstructive surgeries, etc. [1,2,3,4]. Traditional methods for treating bone defects primarily involve autogenous bone grafts and allograft bone grafts [5]. However, autogenous bone grafts have limitations in terms of supply and can lead to complications at the donor site, whereas allograft bone grafts carry risks of immune rejection and infectious diseases [6]. Currently, biodegradable materials represented by biopolymers, ceramics, and metals offer new avenues for bone repair [7,8,9]. Among biomaterials, poly-l-lactic acid (PLLA) is a promising bone graft material owing to its excellent biocompatibility, biodegradability, and processability [10]. After implantation, it degrades into lactic acid through hydrolysis reactions, eventually breaking down into water and carbon dioxide, and can be excreted through normal metabolic pathways. Moreover, it is the first biodegradable medical material approved by the FDA [11]. Unfortunately, standalone PLLA bone scaffolds suffer from limitations such as inadequate biological activity and a lack of antibacterial capabilities [12,13], which greatly restrict their utility in the field of tissue regeneration [14].

Hydroxyapatite (HA) constitutes the main inorganic component of bone tissue, which provides nucleation sites for bone cell calcification and promotes bone conduction and bone induction [15,16]. Moreover, HA is non-toxic, non-hemolytic, and non-irritating in vivo [17,18]. Nano-silver (Ag) possesses broad-spectrum antibacterial abilities, which can bind to pathogenic bacteria cell walls/membranes through electrostatic interactions, and then disrupt the inherent components of bacteria by generating reactive oxygen species (ROS) and interacting with the genetic material of bacteria [19,20,21]. Moreover, Ag is also non-toxic and resistant to resistance [22,23,24]. The combination of these three materials holds great potential for the development of biodegradable bone scaffolds with dual antibacterial and bone-promoting functions. However, these materials exhibit significant disparities in their physicochemical properties, thus leading to their poor compatibility at the interface. Therefore, improving the compatibility between polymers, ceramics, and metals at the interface is a critical challenge.

In situ growth technology is a recent method for fabricating composite materials. It involves using one material as a substrate and creating growth centers for new materials, which achieves a perfect combination of the two at the interface, thereby forming a composite material with a superior performance [25,26,27]. Zeolitic imidazolate frameworks (ZIF-8) are a type of metal–organic framework (MOFs) material formed by the coordination of zinc ions with dimethylimidazole [28,29]. The Zn^2+^ of ZIF-8 can replace the Ca^2+^ in HA through ion exchange, since they have similarities in atomic donor and ion-binding properties [30]. Additionally, the imidazole structure of ZIF-8 can form hydrogen bonds with the carboxyl groups in PLLA. Therefore, ZIF-8 can act as an interface layer to effectively improve the compatibility between HA and PLLA. On the other hand, ZIF-8 can also load nano-Ag into its framework through ion exchange and reduction-fixed methods [31,32]. In this case, Ag loaded in ZIF-8 does not produce interface contact, thus effectively addressing the problem of the weak interface bonding between metals, polymers, and ceramics. Importantly, ZIF-8 can regulate Ag^+^ release through its framework structure, avoiding the cell damage due to sudden Ag^+^ release and achieving the long-term antibacterial efficacy of the bone scaffold.

In this work, a core–shell structure nanoparticle (HA@Ag) was prepared by in situ growing ZIF-8 on the surface of HA and loading Ag into the ZIF-8 framework. Then, HA@Ag nanoparticles were introduced into a PLLA scaffold with interconnected porous structures, aiming to endow the scaffold with good antibacterial activity and osteogenic activity. The composite scaffold was fabricated using selective laser sintering (SLS) technology. The research will focus on the in situ growth mechanism of ZIF-8 on the HA surface, emphasizing the effects of ZIF-8 interface layer enhancement on the mechanical properties, degradation behavior, antibacterial performance, and bone-promoting activity of the composite scaffold.

## 2. Materials and Methods

### 2.1. Synthesis of HA@Ag Nanoparticles

First, 120 mg of nano HA was added into 60 mL of a methanol solution and sonicated for 2 h. Subsequently, 250 mg of zinc nitrate hexahydrate was dissolved in a beaker containing 60 mL of the methanol solution, and then the HA solution was transferred into the above beaker and stirred at 40 °C for 15 min. Next, 280 mg of 2-Methylimidazole was dissolved in 60 mL of the methanol solution and added dropwise to the mixed solution containing nano HA. Finally, the mixed suspension was vigorously stirred for 60 min at room temperature, and then a white powder (HA@ZIF-8) was obtained after three rounds of centrifugation and overnight drying. In order to obtain HA@Ag powder, 200 mg of HA@ZIF-8 powder was put into 80 mL of a silver nitrate solution (0.1 mol/L) for stirring for 1 h, and then 20 mL of a NaBH_4_ solution (0.01 mol/L) was further added into the mixed solution for stirring for 30 h. Finally, the mixed suspension was repeatedly centrifuged 3 times and turned into the expected HA@Ag powder after overnight drying.

The microstructure of the HA@Ag nanoparticles was observed using a transmission electron microscope (TEM, TecnaiG2 20, FEI, Columbia, MD, USA). The phase composition and chemical components of the HA@Ag nanoparticles were analyzed using X-ray diffraction (XRD, Bruker D & Advance, Luken, Germany) and Fourier transform infrared spectroscopy (FTIR, Tianjin Gang Dong Technology Co., Ltd., Tianjin, China), respectively. The element orbital on the bond energy of the HA@Ag nanoparticles was investigated using X-ray photoelectron spectroscopy (XPS, Thermo-VG Scientific Ltd., Waltham, MA, USA).

### 2.2. Fabrication of PLLA/HA@Ag Bone Scaffold

The PLLA/HA@Ag porous bone scaffold was prepared via an independently designed SLS device [33,34]. Before preparation, PLLA and HA@Ag powder usually need to undergo drying at 45 °C overnight. The PLLA and HA@Ag powder were mixed in a certain proportion, stirred uniformly, then poured into alcohol for ultrasonic dispersion; next, the mixed powder was obtained after centrifugation 3 times (8000 rpm, 5 min) and vacuum drying (50 °C, 24 h). Finally, the mixed powder was used as a raw material to fabricate the PLLA/HA@Ag porous bone scaffold. The process parameters of the SLS system were as follows: a scan speed of 6000 mm/s, a laser power of 4 W, and a layer thickness of 0.1 mm. Similarly, the PLLA and PLLA/HA porous bone scaffold were also prepared using the above process parameters. The surface morphology of the scaffold was studied using scanning electron microscopy (SEM, S5000, Hitachi, Tokyo, Japen) and an electronic camera (Canon Kiss X3, Canon Inc., Tokyo, Japan). The phase composition and chemical components of the scaffold were also detected using XRD and FTIR.

### 2.3. Mechanical Property

The mechanical properties of the PLLA/HA@Ag scaffold were investigated through tensile and compressive tests. For the tensile test, dumbbell-shaped specimens were prepared via the SLS system, and their sizes were accordance with the standard of non-metallic stretching splines (ISO 604:2002 standard). As for the compressive test, a porous scaffold with a size of 3 × 3 × 5 mm^3^ was used as the test sample. Then, these samples were placed on the workbench of a universal testing machine (E44.304, Meters Industrial Systems (China) Co., Ltd, Shanghai, China) for testing, in which the loading rate was set as 1 mm/min. Finally, the test dates of the mechanical properties for the scaffold were collected. Every experiment was repeated five times. The stretched section of the scaffold was studied using SEM observation, aiming to analyze the interface bond between the PLLA and HA@Ag.

### 2.4. In Vitro Degradation Experiment

The in vitro degradation behavior of the scaffold was investigated through a soak test in a phosphate-buffered saline (PBS) solution. Specially, the scaffold was immersed in the PBS solution (at 37 °C, pH 7.4). The mass to volume ratio was 0.1 g/mm^3^. After immersion for a specific time (1, 2, 3, and 4 weeks), the scaffolds were dried at 45 °C for 4 h, aiming to eliminate the mass caused by the absorption solution. Finally, the degradation rate of the scaffolds was determined via calculating the weight of the scaffold before and after the soaking. Moreover, the pH change of the soaking solution and the Ag^+^ concentration in the soaking solution were detected using a PH meter (Metes industrial systems Co., Ltd., Anyang, Republic of Korea) and an inductively coupled plasma mass spectrometer (ICP-MS, Plasma 3000, NCS Testing Technology Co., Ltd., Beijing, China), respectively.

### 2.5. In Vitro Antibacterial Experiment

*Staphylococcus aureus* (*S. aureus*, ATCC 49775) was used as infectious bacteria to estimate the antibacterial activity of the PLLA/HA@Ag scaffolds. The antibacterial ratio of the scaffold was evaluated using plate colony-counting methods. In detail, prior to the test, all the scaffolds were sterilized via immersion in ethanol (a concentration of 75%) and UV light for 30 min. Next, the scaffolds and *S. aureus* solution were co-cultured on a glass garden at 37 °C for 24 h, and then the bacterial–scaffold solution was diluted to 1 × 10^9^ CFU/mL. Subsequently, 30 mL of the diluent was placed in Petri dishes containing LB agar and cultured for 12 h at 37 °C. Eventually, images of the bacterial colony were observed using the digital camera, and the corresponding number was measured using the Image J software (Image-J v1.44, NIH Image, Bethesda, MD, USA). The antibacterial ratio was calculated according to the formula:Antibacterial rate (%) = [1 − (CFU sample/CFU control)] × 100%
where CFU control indicates the living bacteria number without any treatment and CFU sample indicates the live bacteria number in the presence or absence of NIR irradiation. Moreover, the bacterial adhesion of the scaffold was investigated. In detail, the scaffold samples were inserted into a centrifuge tube containing 1 × 10^6^ CFUs/mL and incubated at 37 °C for 12 h. The bacterial morphology on the scaffold was also studied using the SEM device. Then, the scaffolds were gently washed two times with PBS to remove the loosely adherent bacteria. Subsequently, ultrasonic cleaning was carried out to remove the adherent bacteria on the scaffold. Next, the collected solutions were subjected to a 106-fold dilution process, and were then plated onto the Petri dishes for incubation at 37 °C for 24 h. Finally, the ultimate CFUs were the number of colonies multiplied by the dilution ratio. The CFUs of each group were normalized to the counts from the PLLA group. The bacterial morphology on the scaffold was also studied using the SEM device. To explore the antibacterial mechanism of the scaffold, a test of the bacterial membrane and an ROS test were carried out, which was because the outer membrane test determined the ability of the metal ions in the scaffold to destroy the bacterial membrane, while the ROS test verified whether the scaffold had the ability to generate ROS. In short, the ROS level and bacterial nuclei of the scaffolds were characterized using 2′,7′-dichlorodihydrofluorescein diacetate (DCFH-DA).

### 2.6. In Vitro Cell Culture Tests

The mBMSCs (derived from Procell Life Science & Technology Co., Ltd., Wuhan, China) were cultured in a 5% CO_2_ humidified incubator at 37 °C using Dulbecco’s modified Eagle’s medium (DMEM, Invitrogen, New York, NY, USA) supplemented with 10% fetal bovine serum (FBS, Gibco, Grand Island, NY, USA), 100 U/mL of penicillin, and 100 µg/mL of streptomycin. The culture medium was replenished as needed. The scaffolds were co-cultured with human skin fibroblasts at a density of 1 × 104 cells per well, maintained at 37 °C for 1, 4, and 7 days. The culture medium was refreshed once daily. After the specified culture period, the co-cultured composites were carefully removed and washed twice with PBS. Subsequently, the composites were subjected to a 30 min staining procedure using calcein-AM and propidium iodide (PI) at 37 °C. Finally, the cellular morphology on the scaffolds was examined using fluorescence microscopy.

To assess the BMSC proliferation on the scaffold, we employed the cell counting kit-8 assay (CCK-8, Beyotime, Nangtong, China) and Calcein-AM/PI Double Stain Kit (Beyotime, Bejing, China), following previously established protocols. For the evaluation of the mBMSC osteogenic differentiation on the scaffold at 4 and 7 days, we measured the alkaline phosphatase (ALP) activity using a bicinchoninic acid protein assay kit (Jiancheng Co., Beijing, China). The optical density (OD) values were determined at 520 nm using a microplate reader. Furthermore, we assessed the degree of mineralization in the mBMSCs via alizarin red staining at 7 and 10 days using 0.1% alizarin red (pH 4.2, Soledad Bao Tech., Nangtong, China) and 10% hexadecyl pyridinium chloride monohydrate (Sigma–Aldrich, St. Louis, MO, USA). The staining effects were examined under an optical microscope, and the OD values of calcium were measured at 550 nm using an enzyme-labeling instrument.

### 2.7. Statistical Analysis

Data are presented as the mean ± standard deviation (n ≥ 3). Odds ratios (ORs) and 95% confidence intervals (CIs) were calculated; two-tailed tests were used to determine statistical significance and a *p* value of ≤0.05 was considered to be significant.

## 3. Results and Discussion

### 3.1. Characterization of HA@Ag Nanoparticles

The synthesis process of the HA@Ag nanoparticles is illustrated in Figure 1a. Briefly, we first utilized the easy functionalization feature of MOF materials to encapsulate the HA with the ZIF-8. Afterwards, Ag^+^ was loaded into the framework structure of ZIF-8 via the ion exchange reaction. Then, the reducing agent NaBH_4_ was used to deoxidize Ag^+^ into Ag, thus obtaining the expected core–shell-structured HA@Ag nanoparticles. To verify this core–shell structure, the morphologies of HA@Ag were investigated using TEM observation, as shown in Figure 1b–b_4_. Evidently, the HA nanoparticles exhibited a block shape with an average size of 65 nm (Figure 1b). As a comparison, HA was covered with a wing-like viscous layer, and some fine nanoparticles grew in the surface of the coated layer (Figure 1b). Based on previous work, the wing-like viscous layer was coated ZIF-8 [32]. High-resolution TEM observation showed that these fine nanoparticles presented a subglobose shape with average size of ~8.5 nm and possessed clear lattice stripes. Determined using a Fourier transform analysis, the average lattice fringe spacing of the fine nanoparticles was 0.243 nm, which was assigned to the (111) plane of the Ag crystals [35]. Moreover, diffraction ring images revealed that the HA@Ag possessed obivious crystal morphology, thus further confirming the successful synthesis of the Ag nanoparticles.

The XRD patterns of the HA and HA@Ag nanoparticles were detected, as shown in Figure 1c. Evidently, three typical characteristic peaks located at 25.9°, 31.9°, and 46.7° appeared on the HA patterns, which corresponded to its (002), (211), and (222) planes, respectively [36,37]. As for HA@Ag, three remarkable peaks located at 38.1°, 44.2°, and 64.5° were observed, which were ascribed to the (111), (200), and (220) of planes of Ag, respectively [38]. Moreover, the halo peak located at 21.1° might have been caused by the ZIF-8 crystals, since the ZIF-8 acted as a sacrificial phase to provide the ion exchange condition of Ag^+^ in the whole synthesis process. Notedly, the characteristic peak of HA disappeared on the XRD patterns, which was due to the shielding effect of the coated ZIF-8 and in situ grown Ag nanoparticles. In summary, the XRD results proved the viewpoint described by the SEM analysis above, referring to the in situ growth of Ag on the surface of HA.

The FTIR spectrums of HA, HA@ZIF, and HA@Ag were studied, as shown in Figure 2a. Evidently, the HA exhibited two peaks located at 3482 and 1045 cm^–1^, which corresponded to the stretching of the hydroxyl group and phosphate group, respectively [39]. After the surface functionalization with ZIF-8, there were two new peaks located at 2821 and 757 cm^–1^, which were assigned to the methoxy group and imidazole ring of ZIF-8, respectively [40]. As for the HA@Ag nanoparticle, the peak caused by the hydroxyl group (3460 cm^–1^) appeared on the FTIR spectra and the characteristic peaks of ZIF-8 disappeared, indicating that the hydroxyl group involved the reduction of Ag^+^ and the structure of ZIF-8 was destroyed in the synthesis process [41]. To explore the concrete combining form between the HA and Ag nanoparticles, the XPS spectra of HA@Ag were investigated, as shown in Figure 2b. It was observed from the total spectrum that each test group displayed unique characteristic peaks, such as Zn bonding orbitals of the HA@ZIF group and Ag bonding orbitals of the HA@Ag group. Furthermore, the high-resolution spectrums showed that the P2p orbitals of HA exhibited two obvious peaks located at 133.45 and 132.98 eV, which were attributed to P2p 3/2 and P2p 1/2, respectively. After the surface functionalization with ZIF-8, the P2p orbitals could be subdivided into three obvious peaks located at 139.9, 138.7, and 134.0 eV, which were assigned to P2p 3/2 and P2p 1/2, respectively. In addition, the Zn2p orbitals exhibited two typical peaks located at 133.5 and 132.98 eV, which corresponded to Zn 2p2/3 and Zn 2p1/2 of ZIF-8. The results effectively proved that Zn^2+^ could replace the Ca^2+^ in HA via ion exchange, and then grew into a ZIF-8layer via a self-assembly reaction. After the incorporation of the Ag element, it was seen that the Ag3d orbitals exhibited two peaks located at 373.1 and 367.2 eV, which represented signatures of the metallic Ag 3d3/2 and 3d5/2 states, respectively. In short, Ag^+^ first entered the ZIF-8 framework through ion exchange, and was then restored to nano Ag through the action of reducing agents (Figure 2c).

### 3.2. Microstructure and Mechanical Properties of the Scaffold

As is well known, the structure design of a scaffold is meaningful for the bone repair process. The bone scaffold requires a stable mechanical performance to meet the in-body requirements, while also necessitating an interconnected porous structure to facilitate cell adhesion, growth, the efficient transport of nutrients, and waste removal. Herein, a representative PLLA/HA bone scaffold with an interconnected porous structure was prepared using SLS methods. As shown in Figure 3a, the morphology of the scaffold was almost consistent with the designed structure. Furthermore, it was seen from the SEM images that its porous size was approximately 720 μm, which satisfied the use requirement (the range of 350–800 μm) [42]. Significantly, the scaffolds exhibited a well-forming quality and dimensional accuracy. The brittle fracture surface of the scaffold was observed to study the dispersion state of the nanoparticles in the PLLA matrix. As shown in Figure 3c, the surface of the PLLA scaffold was very smooth, whereas some obvious nanoparticles appeared on the surfaces of the PLLA/HA and PLLA/HA@Ag scaffolds. Further observation found that the majority of nanoparticles in the PLLA matrix were agglomerated, which was due to the huge van der Waals force and specific surface area between the HA nanoparticles. As for the PLLA/HA@Ag scaffold, these nanoparticles were relatively evenly distributed on the surface of the substrate, indicating that the affinity between the HA@Ag particles was lower than that of the HA nanoparticles. According to our previous work, ZIF-8 possessed a good interface compatibility with the PLLA matrix, which further promoted the dispersion of nanoparticles in the matrix.

The mechanical properties of the scaffold seriously affect bone repair, since they provide a major support effect during the initial stage of implantation. In this work, the compressive and tensile strength of the scaffold were evaluated using a universal mechanical testing machine. Figure 3d shows the compressive stress–strain curves of the scaffolds. It was seen that all the scaffolds showed a linear stress–strain relationship, followed by a sudden mechanical fracture, which was attributed to the inherent brittleness of PLLA. The compressive strengths of the scaffolds were calculated from their stress–strain curves, as shown in Figure 3e. The compressive strength of PLLA was 15.1 MPa, whereas the compressive strengths of PLLA/HA and PLLA/HA@Ag were 18.8 and 20.2 MPa, which were increased by 24.5% and 33.7%, respectively. To the best of our knowledge, the compressive strength of PLLA/HA@Ag was similar to that of cancellous bones (approximately 20 MPa) [43,44], making the scaffold have a great potential for application in bone repair. Notably, the compressive strength of PLLA/HA@Ag was higher than that of PLLA/HA, which might be ascribed to the uniform dispersion and good interface bonding of nanoparticles. The tensile tests showed that the tensile strengths of PLLA/HA and PLLA/HA@Ag were significantly higher than that of the PLLA scaffold, which indicated that the incorporated nanoparticles could serve as reinforcing phases to enhance the mechanical properties of scaffolds. Moreover, compared to PLLA/HA, the tensile strength of the PLLA/HA@Ag scaffolds was increased from 15.8 to 17.2 MPa, which was an increase of 8.9%.

To further study the interfacial compatibility between the nanoparticles and PLLA matrix, the tensile fracture surface of the scaffold was observed, as shown in Figure 4a. Evidently, the tensile fracture surface of PLLA was smooth and flat, which conformed to its brittleness characteristic. As a comparison, PLLA/HA and PLLA/HA@Ag exhibited some ductile tearing fractures with clear pulled filaments. Further observation found that abundant HA nanoparticles in the PLLA/HA scaffold dropped out of the matrix and were exposed on the fracture surface. In contrast, the HA@Ag nanoparticles were well embedded in the pulled filaments, which indicated that there was great interfacial adhesion between HA@Ag and PLLA. As we know, PLLA, HA, and Ag belong to polymers, ceramics, and metal, which makes it difficult for them to form good interface bonds, respectively. In this work, ZIF-8 acted as an interface phase to promote their interface compatibility (Figure 4b). In detail, ZIF-8 grown in situ on the surface of HA could form a good interfacial compatibility with PLLA through interaction between the carboxyl group of PLLA and the amino group of ZIF-8. Meanwhile, Ag nanoparticles were encapsulated in the skeleton of ZIF-8, thereby avoiding the interface bonding. Critical evidence included the relatively uniform dispersion of HA@Ag in the PLLA matrix and the obvious pulled filaments on the tensile fracture surface of PLLA/HA@Ag.

### 3.3. Degradation Behavior of the Scaffold

For bone implants, it is necessary to understand the degradation behavior of scaffolds. The weight loss of the scaffolds after immersion for 28 days was measured, as shown in Figure 5a. Evidently, the degradation rate of the PLLA scaffold was the lowest compared to other groups. After the incorporation of HA, the weight loss of the scaffold after immersion for 28 days increased from 6.73 to 8.67%, which indicated that HA could accelerate PLLA degradation. This might be ascribed to the fact that HA possesses hydrophilic properties, which could promote more water molecules to invade the interior of the scaffold, thereby accelerating the hydrolysis of PLLA. As a comparison, the weight loss of the PLLA/HA@Ag scaffold was 8.56%, and its degradation rate was increased in the last two weeks, which was due to the pH-responsive dissolution of ZIF-8. pH changes in the immersion solution for all scaffolds were also detected, as shown in Figure 5b. Due to the acidic degradation products of PLLA, the pH value decreased from 7.4 to 7.02 after 28 days of soaking. For the PLLA/HA scaffold, the pH value exhibited a slowly declining trend compared to the PLLA scaffold, which was due to the fact that HA hydrolysis is alkaline, which neutralized the acidic products to some extent. As for PLLA/HA@ZIF-8, the pH value decreased from 7.4 to 6.92 after 28 days of soaking, which was due to the increased degradation rate of the scaffold.

The Ag^+^ concentration of the immersion solution for PLLA/HA@ZIF-8 was also studied, as shown in Figure 5c. Clearly, the Ag^+^ concentration increased with an extension of the immersion time, and the concentration was 0.56 ppm after immersion for 28 days, which met the release requirements of Ag^+^ in vivo (<0.8 ppm) [45]. The results revealed that HA@Ag effectively slowed down the Ag release, thus avoiding the cytotoxicity caused by its sudden release. The degradation morphologies of the scaffolds are shown in Figure 5d. A few shallow pits were observed after immersing the PLLA scaffold for 28 days, whereas more and more pits appeared on the surface of PLLA/HA. It is worth noting that the shallow pits were uniformly distributed on the PLLA/HA@Ag surface after 28 days’ immersion, which might be attributed to the uniform distribution of HA@Ag in the PLLA matrix. The results revealed that the PLLA/HA@Ag scaffold exhibited uniform degradation behavior.

### 3.4. Antibacterial Activity of the Scaffold

Endowing a scaffold with strong antibacterial activity is a crucial pathway to ensuring bone regeneration. In this study, the in vitro antibacterial activity of the scaffold was analyzed using Gram-positive *S. aureus* and the control group was a bacterial solution without scaffold treatment. As shown in Figure 6a, the colony numbers of the PLLA and PLLA/HA groups were almost the same as the control group, indicating that PLLA and PLLA/HA had no antibacterial activity. As a comparison, only a few bacteria appeared on the orifice plate of the PLLA/HA@Ag group, which implied that the HA@Ag nanoparticles effectively endowed the scaffold with excellent antibacterial activity. The results confirmed that the PLLA/HA@Ag scaffold could effectively kill bacteria. The antibacterial rate of the scaffold was measured using the bacterial planking results, as shown in Figure 6b. It was seen that the antibacterial rates of the PLLA and PLLA/HA groups were similar to that of the control group. In contrast, the antibacterial rate of PLLA/HA@Ag reached 92.6%, which indicated that the scaffold possessed excellent antibacterial activity. Upon examining the bacterial morphology on the scaffold, it was observed that the bacteria on the PLLA/HA@Ag scaffold exhibited significant deformations, while those on the the PLLA/HA and PLLA scaffolds retained a regular spherical shape (Figure 6c). Similarly, the bacterial adhesion assay showed that PLLA/HA@Ag could effectively resist bacterial invasion as compared to the PLLA scaffold (Figure 6d).

To investigate the antibacterial mechanism of the PLLA/HA@Ag scaffold, a series of experiments, including antibacterial membrane and ROS tests, were carried out, as shown in Figure 6e,f. The fluorescence microscopy images for studying the inner membrane permeability are shown in Figure 6e. DAPI can stain both live and dead cells, whereas PI only stains dead cells [46]. Evidently, the majority of cells for the control group and PLLA and PLLA/HA groups were living cells. As for the PLLA/HA@Ag group, a certain number of dead cells were observed and their corresponding positions overlapped with those of corresponding live cells (merge images), indicating that HA@Ag could damage the inner membrane. As shown in Figure 6f, the OD values of the control group and PLLA and PLLA/HA groups were ~0.28, 0.29, and 0.32, respectively. As a comparison, the OD value of PLLA/HA@Ag was 0.56, which revealed that the scaffold possessed an ability to generate ROS. According to the above results, the PLLA/HA@Ag scaffold could effectively inactivate bacteria via the destruction of bacterial membranes and production of ROS. The antibacterial function of the scaffold was attributed to the Ag component of HA@Ag. In detail, the outer membrane of the bacteria was first damaged by electrostatic action between Ag^+^ and electrostatic action, and then the intrinsic composition of the bacteria was hurt by ROS generation and the interaction between silver and the genetic material of the bacteria.

### 3.5. Cell Response of the Scaffold

As is well known, biocompatibility is a fundamental indicator for determining whether implants can be applied in clinical practice. In this work, stem cells were used as experimental cells. The biocompatibility of the cells cultured on the scaffolds for 4 and 7 days was studied using cell viability staining, as shown in Figure 7a. Evidently, the cell number seeded on all the scaffolds was increased by the extension of the cultivation cycle. Additionally, almost no red cells appeared after culturing for 7 days, which indicated all the scaffolds exhibited a good biocompatibility. Notedly, the cell numbers seeded on PLLA/HA and PLLA/HA@Ag were more than that of the PLLA group, suggesting that they provided a more comfortable environment for cell growth. To further quantitatively evaluate the proliferation of cells on the scaffolds, the cells cultured for 4 and 7 days were measured using a CCK-8 assay (Figure 7b). In general, the OD value is positively correlated with the number of cells [47]. It was seen that the OD value for the PLLA/HA group was highest after culturing for 4 days, whereas there was almost no difference between the PLLA and PLLA/HA groups. After culturing for 7 days, the OD values for PLLA/HA and PLLA/HA@Ag were higher than that of the PLLA group. It was found that the HA components of the PLLA/HA scaffold were the main factors driving cell growth, whereas the incorporation of Ag did not further promote cell growth. The cell viability of the cells on the scaffolds was measured, as shown in Figure 7c. It was found that the viability of the PLLA/HA@Ag scaffold was higher than 95 percent, indicating that the scaffold was a potential implant material.

For bone scaffolds, it is also important to evaluate their impact on cell osteogenic differentiation. ALP is a marker for the early differentiation of osteoblasts, and its expression level reflects the process of bone formation. Hence, the osteogenic differentiation ability of the cells seeded on the scaffold was investigated via ALP staining, as shown in Figure 8a. In general, the deeper the staining, the better the osteogenic differentiation. Clearly, the PLLA group exhibited a certain effect on cell differentiation, which was ascribed to the extracellular matrix action of PLLA [48]. For the PLLA/HA@Ag group, the staining depth of the cells was significantly higher than that for the PLLA group, which indicated that the PLLA/HA@Ag group could promote the osteogenic differentiation of cells. The results were due to the fact that HA has a similar mineral composition to natural bone tissue and an excellent biocompatibility, which could facilitate bone regeneration via interactions with surrounding bone tissue. Further, the quantitative results of the ALP staining also showed that the PLLA/HA@Ag group exhibited a more positive effect on cellular osteogenic differentiation compared to that of the PLLA group, with evidence of increasing the value of ALP activity (Figure 8b).

Evaluating calcium deposition can help to understand the growth of bone cells, as it is one of the markers of osteocyte maturation [49,50]. It was known that alizarin red could react with calcium to form a red complex, and its areas were positively correlated with the degree of cell differentiation. Herein, the calcium deposition of the cells cultured on the scaffolds was explored using alizarin red staining, as shown in Figure 8c. Evidently, with an increased culture time, the red region of the cells on all the scaffolds expanded to varying degrees, and the corresponding color also tended towards a deep red. The results indicated that calcium deposition is a necessary process for cell growth and differentiation. Significantly, after culturing for 10 days, the cells on PLLA/HA@Ag exhibited a large area of red as compared to the PLLA group, indicating that the incorporation of HA@Ag nanoparticles promoted calcium deposition through a chelation effect [51]. A quantitative analysis also further showed that the OD value for PLLA/HA@Ag increased from 0.15 to 0.62, whereas the OD value for the PLLA group increased from 0.15 to 0.62 after culturing for 10 days. Based on the above cell results, it was clear that the PLLA/HA@Ag scaffold not only possessed a good biocompatibility, but also effectively promoted cell proliferation and differentiation.

## 4. Conclusions

In this work, a core–shell-structured nanoparticle (HA@Ag) was prepared and then introduced into a PLLA scaffold to endow the scaffold with good antibacterial activity and osteogenic activity. The results showed that the antibacterial rate of the composite scaffold reached 92.6% against Gram-positive *S. aureus*, and the scaffold effectively promoted cell proliferation and differentiation. Significantly, in this study, ZIF-8 acted as an interface layer to improve the interface bonding between the PLLA matrix and nanofiller. Moreover, the incorporated HA@Ag also enhanced the mechanical strength and degradation rate of the PLLA, which expands its application in the tissue regeneration field.

## Figures and Tables

**Figure 1 nanomaterials-13-02828-f001:**
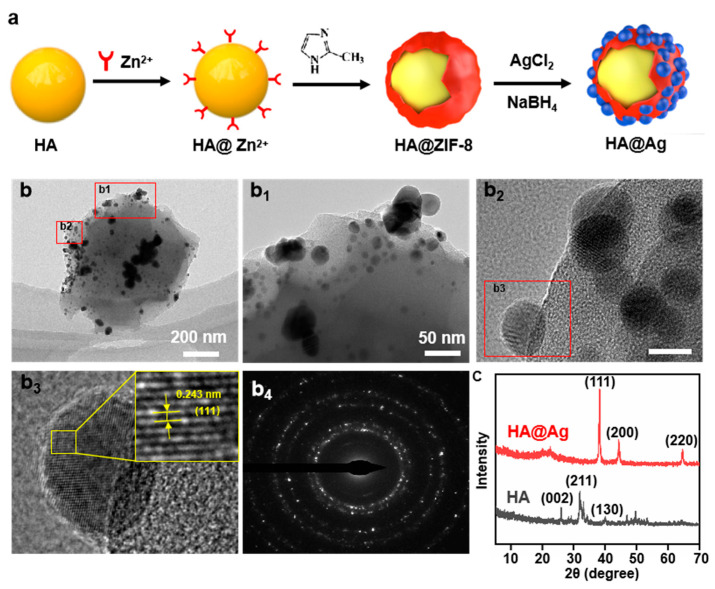
(**a**) The synthesis process of HA@Ag nanoparticles; TEM images of (**b**) HA and HA@Ag nanoparticles; (**b_1_**–**b_3_**) high-resolution TEM images and (**b_4_**) diffraction ring images of HA@Ag nanoparticles; and (**c**) XRD patterns of HA and HA@Ag.

**Figure 2 nanomaterials-13-02828-f002:**
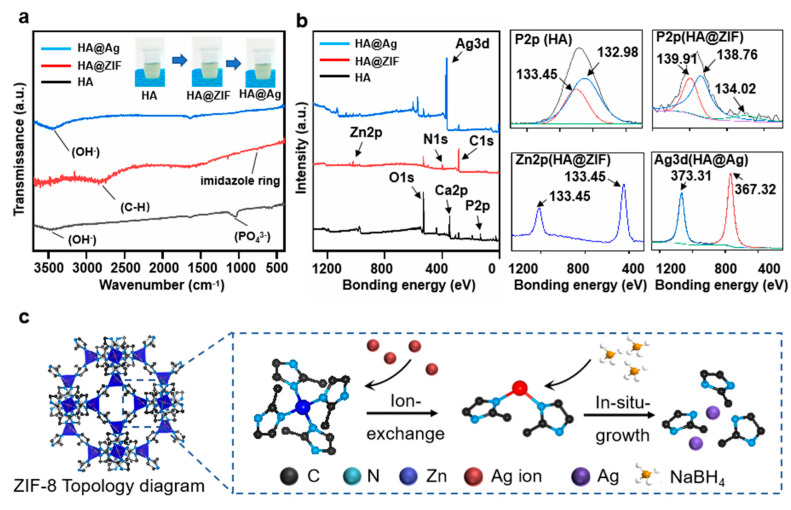
The (**a**) FTIR and (**b**) XPS spectrums of HA, HA@ZIF, and HA@Ag; and (**c**) the schematic diagram for ZIF-8-derived nano-Ag via ion exchange.

**Figure 3 nanomaterials-13-02828-f003:**
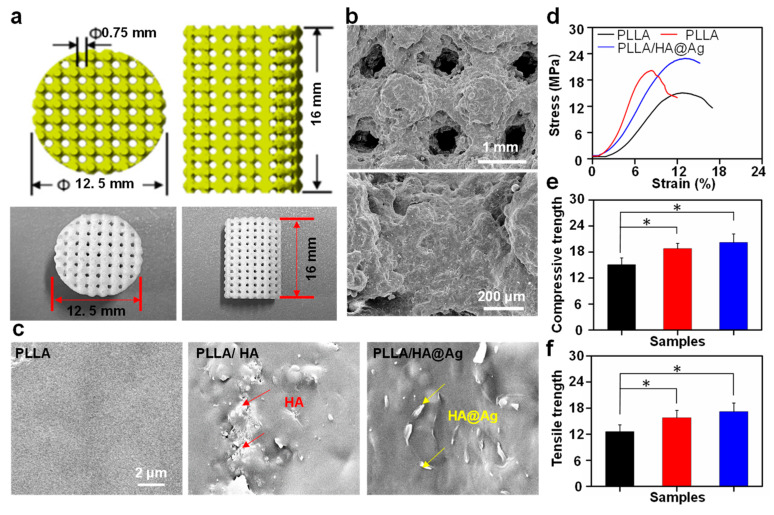
The (**a**) design and actual scaffold structure of PLLA/HA scaffold; (**b**) morphology of PLLA/HA scaffold; (**c**) the brittle fracture surface of all scaffolds; (**d**,**e**) the compressive properties of the scaffolds; and (**f**) the tensile strength of the scaffolds. The * represents the *p*-value less than 0.05.

**Figure 4 nanomaterials-13-02828-f004:**
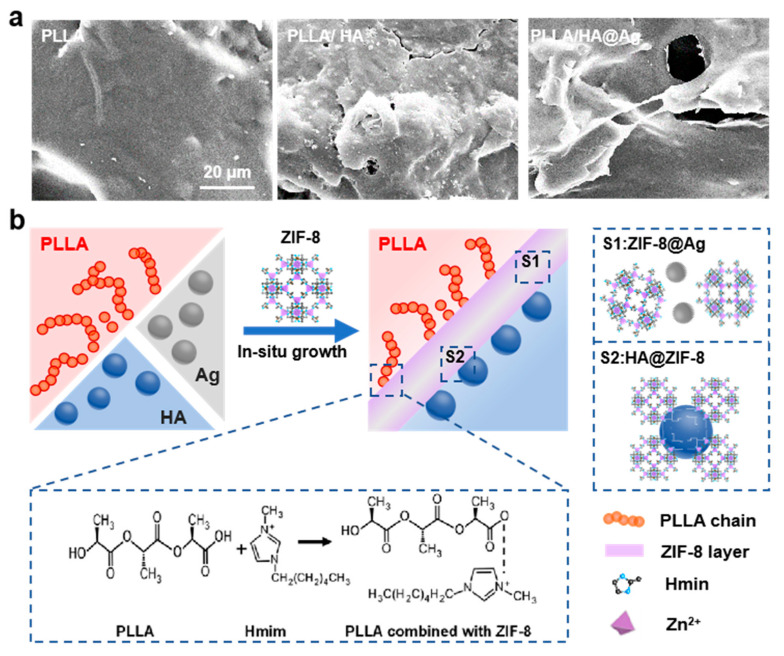
(**a**) Tensile fracture surface of the scaffold; and (**b**) interface compatibility mechanism between of PLLA matrix and HA@Ag nanoparticle.

**Figure 5 nanomaterials-13-02828-f005:**
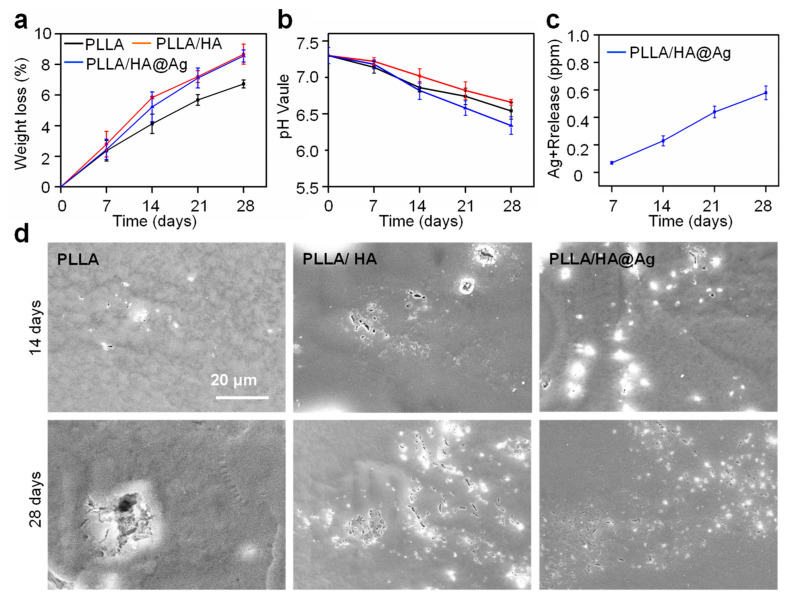
(**a**) Weight loss of the scaffolds; (**b**) the change in pH value in immersion for all scaffolds; (**c**) cumulative release of Ag^+^ concentration for PLLA/HA@Ag; and (**d**) degradation morphologies of the scaffolds.

**Figure 6 nanomaterials-13-02828-f006:**
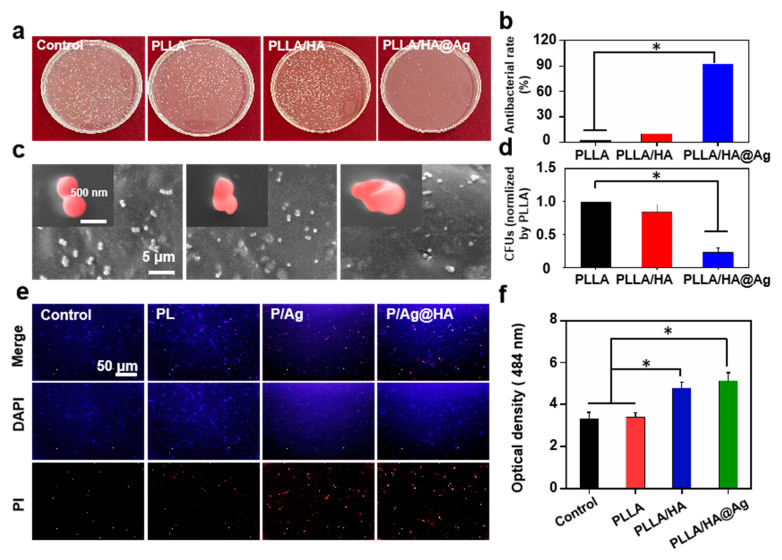
(**a**) Colony testing; (**b**) antibacterial rates of the scaffolds; (**c**) bacterial morphology on the scaffolds; (**d**) bacterial adhesion of the scaffold; (**e**) stained image of the inner membrane permeation of Staphylococcus aureus; and (**f**) effect of the scaffold on the formation of ROS. The * represents the *p*-value less than 0.05.

**Figure 7 nanomaterials-13-02828-f007:**
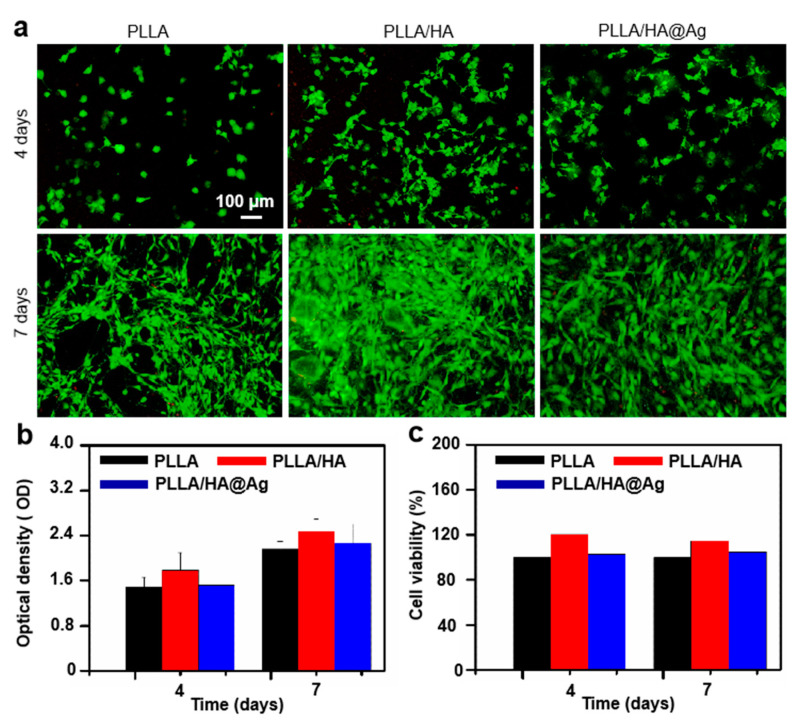
(**a**) Cell fluorescence tests; (**b**) CCK-8 assay; and (**c**) cell viability of stem cells cultured on PLLA, PLLA/HA, and PLLA/HA@Ag scaffolds.

**Figure 8 nanomaterials-13-02828-f008:**
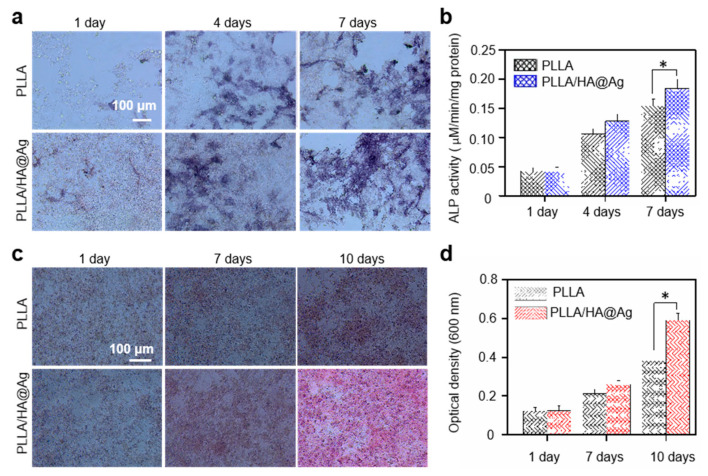
(**a**) ALP staining images and (**b**) corresponding ALP activity of cells on the scaffold; (**c**) Alizarin red staining; and (**d**) corresponding quantitative analysis after cultured for 7 and 10 days. The * represents the *p*-value less than 0.05.

## Data Availability

Data are available upon request.

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
