# Peer review of "Zeolitic Imidazolate Frameworks Serve as an Interface Layer for Designing Bifunctional Bone Scaffolds with Antibacterial and Osteogenic Performance"

_nanomaterials, 2023, doi:10.3390/nano13212828_

Round 1
Reviewer 1 Report
Presented paper is devoted to the formation of bone scaffold using nanoparticles. The topic is relevant. However, there are several comments to it.
There are a huge number of small grammar mistakes and misprint: lines 125, 134, 194, 195, etc. Please double check.
Lines 54, 55. Please provide more supporting papers for this sentence, since there are a lot of articles, in which the toxicity of Ag postulated.
Line 66. Please indicate what means MOF for the convenience of the reader.
2. Materials and Methods. Please provide manufacturer of all used chemicals, equipment, and software.
Lines 136, 137. What is the probability that during soaking the samples will sorb the solution, thereby distorting the data?
Line 144. Were samples washed after ethanol?
Line 152. Please present this equation in more appropriable form.
Line 160. Please clarify in which part of article the using E. coli is presented.
Line 187. Please add more information about statistics. For example, what method, software were used, etc.
Fig. 1f. Please delete “a.u.” since there are no numerical values.
Fig. 3e, f. Presented values must be compared with bone for better understanding of obtained results. It is not clear, which result for which sample is.
Lines 284–286. Above in the text, the authors talk about particle agglomeration, but here about uniform distribution. Please clarify.
Fig. 4b. Please clarify what type of bonding is presented between C and H (?) (see “Combinaton”). PLLA formula is “squished”.
Fig. 5c. Please indicate that release is cumulative.
Fig. 7c. Please provide bars.
There are a huge number of small grammar mistakes and misprint: lines 125, 134, 194, 195, etc. Please double check.
Reviewer 2 Report
Manuscript ID: nanomaterials-2654922
The manuscript entitled “Zeolitic imidazolate frameworks serves as an interface layer to design bifunctional bone scaffold with antibacterial and osteogenic performance” by Jingxi Huang et al. reported the manufacturing of bifunctional bone scaffold with antibacterial and osteogenic properties. The manuscript needs to be really improved, mainly by detailing/enhancing as follows.
In the introduction, the main topics are not explored in depth (e.g. lines 36-42, lines 47-49, lines 54-60, etc.) and the aim of the research is not justified by a comprehensive literature basis. For example, these publications, for example, can be introduced to better answer to this concern:
doi: 10.3389/fendo.2020.00386
doi: 10.3390/antibiotics11040529
doi: 10.1021/acsbiomaterials.2c00140
doi: 10.3390/s22207952.
The degradation assay (lines 131-139) was not executed in simulated body fluids, the in vitro cell culture test (lines 164-186) were not done on osteoblasts, and the antibacterial tests (lines140-163) were not performed by adhesion assays; that all fit better with the aim of the prepared scaffolds, such as human implantable device for orthopedic applications. The Rewiever suggest performing all these types of experiments to improve the specificity of the research. Notably, the results should be included and discussed within the new version of the paper.
The authors provide no discussion section: why? The comparison with literature is a key aspect of a paper and this “lacking” reflects a poor preparation of the manuscript.
Reviewer 3 Report
The paper has been designed well. Here are my minor comments:
1- The particle size of ZIF-8 can be controlled by using a competitive ligand. why did the authors decide to use this particle size for ZIF-8?
2- Clearly the advantages of this material in comparison with your previous work. https://doi.org/10.1007/s42242-021-00130-x
Round 2
Reviewer 1 Report
The authors responded to the comments submitted. I believe that after additional checking of the quality of the English , the paper can be accepted.
The authors responded to the comments submitted. I believe that after additional checking of the quality of the English , the paper can be accepted.
Author Response
The authors responded to the comments submitted. I believe that after additional checking of the quality of the English , the paper can be accepted.
Response:
Thank you very much for your comments. We carefully checked the grammar errors in the manuscript and further improved the quality of English. All the modifications have been highlighted in yellow in the revision and as follows:
Traditional methods for treating bone defects primarily involve the autogenous bone graft and the allograft bone graft.
...
Moreover, Ag is also non-toxic and resistant to resistance.
...
Importantly, ZIF-8 can regulate Ag+ release through its framework structure, avoiding the cell damage due to sudden Ag+ release and achieving long-term antibacterial efficacy of the bone scaffold.
...
Afterwards, Ag+ was loaded into framework structure of ZIF-8 by ion exchange reaction. Then, the reducing agent NaBH4 was used to deoxidize Ag+ into Ag, thus obtaining expectantly core-shell structured HA@Ag nanoparticles.
...
A few shallow pits were observed after immersing the PLLA scaffold for 28 days, whereas more and more pits appeared on the surface of PLLA/HA.
...
To investigate the antibacterial mechanism of PLLA/HA@Ag scaffold, a series of experiments including antibacterial membrane and ROS tests were carried out, as shown in Figs. 6e-f.
...
Moreover, the incorporated HA@Ag also enhanced the mechanical strength and degradation rate of PLLA , which expands its application in the tissue regeneration field.
Reviewer 2 Report
Manuscript ID: nanomaterials-2654922
The manuscript entitled “Zeolitic imidazolate frameworks serves as an interface layer to design bifunctional bone scaffold with antibacterial and osteogenic performance” by Jingxi Huang et al. reported the manufacturing of bifunctional bone scaffold with antibacterial and osteogenic properties.
The authors tried to improve the overall quality of the paper since many modifications were made in the manuscript text, addressing further comments in the letter.
The Reviewer suggests a Moderate editing of English language.
The reviewer suggests to revise by rephrasing some sentences and to corrects typos.
Author Response
The reviewer suggests to revise by rephrasing some sentences and to corrects typos.
Response:
Thank you very much for your comments. We carefully checked the grammar errors in the manuscript and further improved the quality of English. All the modifications have been highlighted in yellow in the revision and as follows:
Traditional methods for treating bone defects primarily involve the autogenous bone graft and the allograft bone graft.
...
Moreover, Ag is also non-toxic and resistant to resistance.
...
Importantly, ZIF-8 can regulate Ag+ release through its framework structure, avoiding the cell damage due to sudden Ag+ release and achieving long-term antibacterial efficacy of the bone scaffold.
...
Afterwards, Ag+ was loaded into framework structure of ZIF-8 by ion exchange reaction. Then, the reducing agent NaBH4 was used to deoxidize Ag+ into Ag, thus obtaining expectantly core-shell structured HA@Ag nanoparticles.
...
A few shallow pits were observed after immersing the PLLA scaffold for 28 days, whereas more and more pits appeared on the surface of PLLA/HA.
...
To investigate the antibacterial mechanism of PLLA/HA@Ag scaffold, a series of experiments including antibacterial membrane and ROS tests were carried out, as shown in Figs. 6e-f.
...
Moreover, the incorporated HA@Ag also enhanced the mechanical strength and degradation rate of PLLA , which expands its application in the tissue regeneration field.